# Dissection of the Complex Transcription and Metabolism Regulation Networks Associated with Maize Resistance to *Ustilago maydis*

**DOI:** 10.3390/genes12111789

**Published:** 2021-11-12

**Authors:** Xinsen Ruan, Liang Ma, Yingying Zhang, Qing Wang, Xiquan Gao

**Affiliations:** 1State Key Laboratory for Crop Genetics and Germplasm Enhancement, Nanjing Agricultural University, Nanjing 210095, China; 2017201073@njau.edu.cn (X.R.); 2019201077@njau.edu.cn (L.M.); 2019101075@njau.edu.cn (Y.Z.); qingwang@njau.edu.cn (Q.W.); 2Jiangsu Collaborative Innovation Center for Modern Crop Production, Nanjing Agricultural University, Nanjing 210095, China; 3College of Agriculture, Nanjing Agricultural University, Nanjing 210095, China; 4Xiangyang Academy of Agricultural Sciences, Xiangyang 441057, China

**Keywords:** maize, resistance, transcriptome, metabolome, *Ustilago maydis*

## Abstract

The biotrophic fungal pathogen *Ustilago maydis* causes common smut in maize, forming tumors on all aerial organs, especially on reproductive organs, leading to significant reduction in yield and quality defects. Resistance to *U. maydis* is thought to be a quantitative trait, likely controlled by many minor gene effects. However, the genes and the underlying complex mechanisms for maize resistance to *U. maydis* remain largely uncharacterized. Here, we conducted comparative transcriptome and metabolome study using a pair of maize lines with contrast resistance to *U. maydis* post-infection. WGCNA of transcriptome profiling reveals that defense response, photosynthesis, and cell cycle are critical processes in maize response to *U. maydis*, and metabolism regulation of glycolysis, amino acids, phenylpropanoid, and reactive oxygen species are closely correlated with defense response. Metabolomic analysis supported that phenylpropanoid and flavonoid biosynthesis was induced upon *U. maydis* infection, and an obviously higher content of shikimic acid, a key compound in glycolysis and aromatic amino acids biosynthesis pathways, was detected in resistant samples. Thus, we propose that complex gene co-expression and metabolism networks related to amino acids and ROS metabolism might contribute to the resistance to corn smut.

## 1. Introduction

Plants have deployed a highly sophisticated innate immune system to perceive potentially dangerous microbes; however, pathogens have also developed strategies to facilitate their own progression on host plants. Therefore, the “arms race” between plants and pathogens is ubiquitous throughout co-evolution history [1]. On one hand, plant can recognize the pathogen-associated molecular patterns (PAMP) of pathogens to subsequently activate PAMP-triggered immunity (PTI) [2]. On the other hand, pathogens could evade PTI through PAMP modification and secreted effectors [3,4,5]. Furthermore, effectors could be perceived by host-resistance (*R*) proteins, such as nucleotide-binding leucine-rich repeats (NB-LRR) proteins, which subsequently activate more robust and rapid defense response, so-called effector-triggered immunity (ETI), often leading to hypersensitive response (HR) or programmed cell death (PCD) in the host [6]. However, the roles of PCD in plant innate immunity largely rely on the living style of colonizing pathogens; e.g., PCD is beneficial to necrotrophic pathogen, but it enables the host to restrict the invasion of biotrophic pathogens. Therefore, to facilitate their pathogenicity, necrotroph and biotroph pathogens have evolved differential strategies to adapt to host immunity.

As one of the top ten plant fungal pathogens, the well-known biotrophic pathogen *U. maydis* causes corn common smut diseases on almost all types of maize organs, resulting in substantial yield loss and quality reduction [7]. Accumulating evidence has shown that *U. maydis* can deploy several strategies to overcome maize immunity, thus initiating the biotrophic interaction with the host [8,9,10]. For example, *U. maydis* effectors, including Jsi1 and Cce1, can suppress the host immune system to promote the virulence [11,12]. Conversely, *U. maydis* infection can also trigger host immune responses, including reactive oxygen species (ROS) production, protease activation, and salicylic acid signaling. Therefore, *U. maydis*–maize interaction is becoming a well-established model to study the interaction between maize and biotrophic pathogens.

Previous works have revealed that multiple maize genes could be involved in defense against *U. maydis*; for instance, the expression of maize *chitinases* was up-regulated in *U. maydis* infected leaves, which subsequently suppressed the fungal growth [13]. Moreover, maize peroxidase POX12 and Papain-like cysteine proteases (PLCPs) were specifically induced, triggering ROS accumulation and salicylic acid related defense signaling, respectively, and thus a defense response to *U. maydis* [14,15]. Furthermore, maize DUF26-domain family proteins were highly induced at an early stage upon *U. maydis* infection, showing antifungal activity, but targeted and suppressed by *U. maydis* effector protein Rsp3 (Repetitive secreted protein 3) [16]. Despite research progress on the interaction between maize and *U. maydis*, host genes and associated resistance strategies deployed by maize to achieve successful defense to *U. maydis* have not been fully identified and characterized.

A previous RNA-seq study showed that numerous maize genes were differently regulated upon *U. maydis* inoculation, and functional categories of different expressed genes (DEGs) were mostly enriched in defense and metabolic pathways [17]. Intriguingly, *U. maydis* infection could also regulate a series of organ-specific genes in maize [10,18]. For example, *U. maydis* effector protein See1 induced maize cell cycle gene expression in leaves, leading to leaf cell division and eventually tumor formation [19]. Furthermore, maize-line-specific genes were also found to be involved in *U. maydis* and maize interaction [20]. Taken together, these findings indicate that the regulation of transcription reprogramming of host genes is critical for the biotrophic infection of *U. maydis*.

In this study, to identify regulatory networks of genes and metabolites associated with resistance to corn smut caused by *U. maydis*, we performed *U. maydis* virulence assay in a panel of maize inbred lines and conducted transcriptome and metabolome analysis using a pair of maize lines with contrasting phenotype. The joint analysis of transcriptomics and metabolomics provided more detailed explanation of the genetic and molecular mechanisms underlying the complex resistance trait of corn common smut at both transcription and metabolism levels.

## 2. Materials and Methods

### 2.1. Plant Materials and Fungal Strain

A maize panel was used consisting of 100 tropical and subtropical varieties, most of which were a gift from Prof. Jianbin Yan at Huazhong Agricultural University. A *U. maydis* strain provided by Prof. Canxin Duan at Chinese Academy of Agricultural Sciences was isolated from infected plants in a corn field of Hebei province, China.

### 2.2. Plant and Fungal Cultivation and Fungal Virulence Assay

The seeds of various maize lines were sown and grown in pots containing potting soil (nutrient soil: vermiculite = 1:1). The seedlings were cultured and maintained in a temperature-controlled greenhouse with a 14/10 h light/dark period, and the light was provided with a photon flux density of 300 μmol·m^−2^·s^−1^, and with temperatures of 28 °C day/20 °C night. *U. maydis* strains were grown in liquid YEPS medium (0.4% yeast extract [*w*/*v*], 0.4% bacto-pepton [*w*/*v*] and 2% sucrose [*w*/*v*]) at 28 °C 200 rpm for 2 days and centrifuged for 20 min at speed of 3000 rpm to remove the medium, and then cell suspension in H_2_O (with 0.01% Tween 20) was adjusted to OD_600_ = 1.0. Suspension was injected with a syringe into the stem of 7-day-old maize seedling, and the disease symptoms of individual plants were scored at 8 days post injection, according to the established classification of disease symptoms [21].

### 2.3. Transcriptome Sequencing and Data Processing

The 7-day-old seedlings of resistant line CML326 and susceptible line GEMS15 were injected with *U. maydis* at 1 h before the light period as described above. Samples of the stem around the injection site were collected at 0 (mock), 1, 2, and 4 days post-infection (dpi) and quickly frozen in liquid nitrogen. Three biological replicates were conducted, and six plants were sampled for each time point. Each subset of samples was pooled and ground in liquid nitrogen, and the RNA was extracted with TRIZOL (Invitrogen, Carlsbad, CA, USA; catalog # 15596-026) by following the manufacturer’s instructions. RNA samples were analyzed on 1% agarose gel electrophoresis, OD260/280 and OD260/230 values of RNA samples were qualified with NanoDrop 2000 (Thermo Fisher Scientific; Waltham, MA, USA), and RIN (RNA Integrity Number) of RNA samples were measured with Agilent 2100 RNA 6000 Nano kit (Agilent Technologies, Santa Clara, CA, USA; catalog # 5067–1511) (Appendix A).

Total RNA was subjected to quality control prior to library construction using Illumina TruSeq RNA library prep kit v2. Subsequently, the RNA sequencing was performed using Illumina HiSeq 2000 (Illumina Inc.; San Diego, CA, USA) at Berry Company, Beijing, China. Approximately 24 million 150 bp paired-end clean reads were obtained per sample.

Hisat2 software version 2.1.0 (Baltimore, MD, USA) [22] was applied to map read pairs to B73 AGPv4.59 reference genome. Stringtie software version 1.3.5 (Baltimore, MD, USA) [23] was applied in assembly with B73 v4.59 reference genome annotation, and gene counts data were processed with R package DESeq2 version 1.31.2 (Heidelberg, Germany) [24]. Gene expression data of 0 day samples were set as control, and DEGs were filtered with the parameters of adjusted *p*-value < 0.05 and absolute log2FoldChange > 1. For gene ontology and KEGG pathway enrichment, R package ClusterProfiler version 3.18.0 (Guangzhou, China) [25] was implemented, and enrichment results were filtered with the parameters of adjusted *p*-value < 0.05.

### 2.4. Weighted Gene Co-Expression Network Analysis

Gene co-expression modules were constructed with R package WGCNA (weighted gene co-expression network analysis) version 1.68 (Los Angeles, CA, USA) [26], for which the soft threshold was set as 14, and a correlation threshold value of 0.75 was applied in the modules’ merging. Genes with module membership over 0.8 were selected as high module membership genes for each module and processed for the enrichment and co-expression study. Furthermore, genes with high co-expression connection within the module were filtered, and co-expression network illustration was conducted with Cytoscape version 3.7.1 (Seattle, WA, USA) [27].

### 2.5. Quantitative Real-Time PCR

Total RNA samples were reverse-transcripted to cDNA using HiScript II Q RT SuperMix kit (Vazyme, catalog # R223-01; Nanjing, China). The relative expression levels of selected genes were detected by quantitative real-time PCR (qRT-PCR) using AceQqPCR SYBR Green Master Mix (Vazyme, catalog # Q111-02; Nanjing, China) and BioRad CFX96 Real-Time PCR Detection System (Bio-Rad Laboratories; Hercules, CA, USA). The melting curves analysis were carried out at the end of PCR cycles, and the relative expression levels of target genes were calculated using the 2^−ΔΔCT^ method [28]. *ZmGAPDH* was used as maize housekeeping gene. Gene specific primers were listed in Appendix A.

### 2.6. Metabolites Quantification and Analysis

A set of leaf samples of mock and infection at 4 dpi were collected for metabolite quantification. The freeze-dried tissue was crushed to fine powder, and 100 mg was extracted with 0.6 mL of 70% aqueous methanol and then filtrated and subjected to UPLC-MS/MS system (UPLC, Shim-pack UFLCSHIMADZU CBM30A system; MS, Applied Biosystems 4500 Q TRAP) for detection and quantification. Metabolites were extracted and identified using Metware database (METWARE, Wuhan, China). Metabolites showing significant accumulation were filtered by R package MetaboAnalystR version 1.01 (Montreal, QC, Canada) [29], with parameters of absolute log2FoldChange > 1 and variable importance in projection (VIP) over 1. Principal component analysis (PCA) was performed by a statistics function within R. All charts were illustrated with R package ggplot2.

## 3. Results

### 3.1. Transcriptome Profile of Resistance and Susceptible Maize Lines upon U. maydis Infection

To identify the maize germplasms resistant to *U. maydis* from a panel consisting of tropical and subtropical lines, we performed *U. maydis* virulence assay on seedlings of 100 lines. The panel displayed great variation of the average disease index, ranging from level 1 to level 6 (Figure 1A, Appendix A). Through more than three biological replications of phenotype assay, two lines showed a stable and contrasting phenotype, with CML326 being resistant, mostly covering level 2 and level 3, whereas GEMS15 was highly susceptible to *U. maydis* infection, covering level 4 and level 5 (Figure 1B,C).

To identify the genes potentially involved in resistance to *U. maydis*, we carried out a comparative transcriptomic analysis using CML326 and GEMS15 lines inoculated with *U. maydis* at 0, 1, 2, and 4 dpi, respectively. The RNA samples displayed high quality, as evidenced by both electrophoresis analysis (Appendix A) and RIN values (Appendix A). A strong correlation was found among three biological replicates (Appendix A). DEGs of maize upon *U. maydis* infection over mock samples were then identified. Overall, more DEGs were found in both lines at 1 dpi and 2 dpi compared to at 4 dpi. Moreover, more numbers of DEGs were detected in resistant line CML326, with 5267 and 5044 up-regulated at 1 dpi and 2 dpi, respectively, while 4569 and 3980 DEGs were detected in the susceptible one GEMS15 at 1 dpi and 2 dpi, respectively. Similarly, more down-regulated DEGs were found in CML326 at both 1 and 2 dpi compared to in GEMS15. However, the number of DEGs between CML326 and GEMS15 at 4 dpi was similar (Figure 1D). DEGs of all time points were then combined and analyzed. In total, 6088 up-regulated and 4670 down-regulated genes were uniquely expressed in resistant line CML326, among which 13 genes were commonly detected for both up- and down-regulated genes. Meanwhile, 5266 and 3459 DEGs were up-regulated and down-regulated in susceptible line GEMS15, respectively, with 4 DEGs sharing both expression patterns. Moreover, 4155 common up-regulated genes in total were detected between two lines, while 1933 and 1111 genes were specifically up-regulated in CML326 and GEMS15, respectively. In the case of down-regulated genes, 2444 common genes were shared between two lines, while 2226 and 1015 DEGs were identified to be specifically down-regulated in CML326 and GEMS15, respectively (Figure 1E). Furthermore, the expression changes of seven randomly selected genes, including *ZmRPS2*, *ZmPK*, *ZmGP8-4*, *ZmNTH*, *ZmAuxTF*, *ZmSBP,* and *ZmCM*, in RNA-seq data were validated with qRT-PCR. The melting curve analysis showed an obvious peak for all genes tested, suggesting a specific amplification of qRT-PCR (Appendix A), and a strong correlation was observed between RNA-seq and qRT-PCR data (Appendix A).

To determine DEGs that might be involved in the resistance to *U. maydis*, Gene Ontology (GO) analysis was enriched from both up-regulated and down-regulated DEGs. GO terms for the Biological Process (BP) in up-regulated genes were mainly enriched in defense response (GO:0006952), response to biotic stimulus (GO:0009607), and response to oxidative stress (GO:0006979). Enriched GO terms for up-regulated genes included heme binding (GO:0020037), carbohydrate binding (GO:0030246), and oxidoreductase activity paired donors (GO:0016705) (Appendix A). Enriched biological process GO-terms for down-regulated DEGs included photosynthesis (GO:0015979), cell cycle (GO:0007049), and pigment metabolic process (GO:0042440), while molecular function GO terms for the down-regulated DEGs were mainly enriched in chlorophyll binding (GO:0016168), microtubule binding (GO:0008017), and catalytic activity, acting on DNA (GO:0140097) (Appendix A). In summary, GO enrichment analysis of DEGs suggested that *U. maydis* could induce the activation of defense response and ROS, and photosynthesis, cell cycle, and primary and secondary metabolism in maize host.

### 3.2. WGCNA Analysis of Function Gene Networks in Maize Response to U. maydis

To further characterize the function gene networks that might be associated with maize resistance to *U. maydis*, WGCNA was deployed to analyze the expression patterns of DEGs. In total, 24,218 DEGs were subjected to WGCNA, and 10 modules were identified based on their co-expression patterns (Figure 2A). Most genes were clustered in five modules, including turquoise, blue, brown, yellow, and green (Figure 2B). Module eigengene analysis showed that major genes in the turquoise module were induced at 1 and 2 dpi in response to *U. maydis*, whereas those in the brown and yellow module were down-regulated at 1 and 2 dpi upon infection with *U. maydis*. However, genes in the blue module did not show obvious changes (Figure 2C). It is noticeable that 351 and 642 genes in the blue module via K-means clustering were specifically highly expressed in GEMS15 and CML326, respectively (Figure 2D), which is consistent with the DEGs count results.

To understand the functions of DEGs identified through WGCNA, GO enrichment analysis was conducted. GO terms in biological process (BP) in turquoise module included defense response (GO:0006952), cellular amino acid metabolic process (GO:0006520), interspecies interaction between organisms (GO:0044419), lipid biosynthetic process (GO:0008610), and reactive oxygen species metabolic process (GO:007259), suggesting that the metabolism regulation in amino acid, lipid, and ROS was highly correlated with defense response of maize to *U. maydis*. Furthermore, DEGs in the brown module were mainly enriched in processes involving photosynthesis (GO:0015979), while those in the yellow module were mainly enriched in processes related to the cell cycle (GO:0007049) (Figure 2E). A series of genes were highly induced in CML326 upon infection with *U. maydis*, among which several genes involved in plant defense to pathogens were significantly strongly up-regulated, such as *100283208* (*Zm00001d014840*, *basic endochitinase 1*), *542243* (*Zm00001d014842*, *class I acidic chitinase*), and those related to ROS regulation, *103635232* (*Zm00001d009349*, *putative respiratory burst oxidase homolog protein H*), *103655132* (*Zm00001d049110*, *Aspartic acid proteinase inhibitor pseudogene*), *100273550* (*Zm00001d017867*, *Aspartyl protease AED1*), and amino acid metabolism regulation *100279999* (*Zm00001d007462*, *tyrosine aminotransferase homolog1*) (Appendix A).

### 3.3. Gene Co-Expression Network of Maize upon Infection with U. maydis

To characterize the gene regulation network of maize inoculated with *U. maydis*, the top 100 genes were filtered based on co-expression patterns in WGCNA, and networks were conducted in turquoise, brown, and yellow modules, respectively. In the turquoise module network, two genes, including *100383079* (*Zm00001d032869*, *putative lectin-like receptor protein kinase family protein*) and *100276695* (*Zm00001d043121*, *Osmotin-like protein OSM34*) that are related to defense response displayed obviously higher co-expression within the module. Moreover, three genes, *542229* (*Zm00001d013098*, *indole-3-acetaldehyde oxidase*), *100192955* (*Zm00001d030613*, *ceramide inositol phosphotransferase 1*), and *100216857* (*Zm00001d017912*, *ASC1-like protein 2*) with stronger co-expression were related to lipid biosynthesis process, and two genes, *100191513* (*Zm00001d023694*, *2-oxoisovalerate dehydrogenase subunit α 2*) and *103627433* (*Zm00001d017276*, *phenylalanine ammonia-lyase*, *PAL*), were annotated on the amino acid biosynthesis process (Figure 3A).

In brown module network, three genes related to porphyrin metabolic process, namely *100191140* (*Zm00001d015366*, camouflage 1), *542554* (*Zm00001d008203*, protoporphyrinogen IX oxidase), and *100193255* (*Zm00001d029027*, ycf54-like protein), and another three genes related to photosynthesis, including *100279224* (*Zm00001d021368*, Tetratricopeptide repeat [TPR]-like superfamily protein), *100280179* (*Zm00001d030638*, PsbP domain-containing protein 1), and *100192838* (*Zm00001d011362*, PsbP domain-containing protein 5), were identified to be co-expressed (Figure 3B). Moreover, two genes related to cell cycle, *542239* (*Zm00001d043158*, β-6 tubulin) and *100192665* (*Zm00001d014885*, nucleic acid binding protein), showed the co-expression pattern in the yellow module network (Figure 3C).

### 3.4. Metabolomic Analysis of Resistant and Susceptible Lines upon U. maydis Infection

To pinpoint the potential metabolites associated with resistance to *U. maydis*, we performed a metabolomic analysis using the above two lines upon infection with *U. maydis*. In total, 505 metabolites were identified from lines CML326 and GEMS15 at 4 dpi and their corresponding mock treatments. The majority of compounds were clustered into flavonoids, phenolic acids, amino acids and their derivatives, lipids, alkaloids, nucleotides and derivatives, and organic acids. Another two clusters, namely lignans/coumarins and terpenoids, contained a small number of compounds (Figure 4A). PCA analysis showed that all compounds were well clustered, and PC1 and PC2 accounted for 33.63% and 22.46% of genetic variation for treatments and pedigrees, respectively, suggesting the strong and specific activation of metabolites between both lines and between infection and control treatment (Figure 4B). Furthermore, the numbers of common and specific differentially accumulated metabolites (DAMs) between two lines were represented in a Venn diagram. There were 127 and 153 metabolites that were significantly increased, whereas 33 and 34 metabolites were significantly decreased in CML326 and GEMS15 upon *U. maydis* infection, respectively (Figure 4C). A total of 101 DAMs were commonly identified in both resistant and susceptible lines upon infection, most of which were secondary metabolites. Among those metabolites, 9 lipids, 4 amino acids and derivatives (Ac-Trp, Ac-Arg, Ac-Asp, L-Pipecolinic acid), 2 vitamins, 1 organic acid, and 1 nucleotide and derivative were up-regulated; on the other hand, 36 flavonoids, 27 phenolic acids, 20 alkaloids, and 1 lignan were down-regulated. Furthermore, 26 DAMs were specifically identified in resistance line, among which there were 7 amino acids and derivatives (Asp, Aspartic acid, Arg, Ser, Lys, Glu and Tyr), 1 lipid, Hexadecyl ethanolamine and O-Phosphorylethanolamine, and 14 secondary metabolites, including 11 flavonoids and 3 phenolic acids, whereas 52 DAMs were found in susceptible line, containing 16 primary metabolites, including 8 lipids, 4 organic acids, and 4 nucleotides and derivatives, as well as 33 secondary metabolites, including 16 phenolic acids, 15 flavonoids, 1 lignan, and 1 alkaloid (Appendix A).

Furthermore, 15 compounds were detected to be commonly decreased between two lines upon infection, among which there were 6 primary metabolites, including 4 nucleotides and derivatives, 1 amino acid and derivative, and 1 saccharide (Glucarate O-Phosphoric acid), and 9 secondary metabolites, including 5 flavonoids, 2 alkaloids, and 2 phenolic acids. Specifically, 18 metabolites were down-regulated in the resistant line, among which 6 organic acids, 2 amino acids and derivatives (His, D-meGly), 2 saccharides (Arabinose, Glucose), 1 lipid, and 1 nucleotide and derivative belonged to primary metabolites, while 4 phenolic acids and 1 alkaloid were secondary metabolites. Moreover, 18 metabolites were specifically down-regulated in the susceptible line, among which primary metabolites included 2 amino acids and derivatives (Pro, phe-Gly), 2 organic acids, and 2 lipids, while secondary metabolites contained 6 flavonoids, 2 phenolic acids, 1 lignan, and 1 terpenoid (Corosolic acid) (Appendix A). Intriguingly, it clearly showed that multiple flavonoid and phenolic acids were accumulated upon *U. maydis* infection, whose contents were specifically enhanced in the susceptible line, whereas seven amino acids and derivatives were specifically detected in the resistance line upon *U. maydis* infection (Figure 4D). In line with this, KEGG pathway enrichment of high module membership genes in turquoise module showed that the biosynthesis of amino acids (zma01230), glycolysis/gluconeogenesis (zma00010), α-linolenic acid metabolism (zma00592), and phenylpropanoid biosynthesis (zma00940) were significantly enriched (Figure 4E).

### 3.5. Shikimic Acid Pathways Associated with Defense Response against U. maydis

To assert the involvement of amino acids biosynthesis in resistance to *U. maydis* identified by metabolomic analysis, we conducted a joint analysis of transcriptome and metabolome data to investigate the association of gene and metabolites of glycolysis, aromatic amino acids biosynthesis, phenylpropanoid biosynthesis, and flavonoid biosynthesis pathways in both lines in response to *U. maydis* infection.

Among the metabolites of glycolysis pathway, the amount of sucrose remained unchanged between mock and *U. maydis* inoculated samples in both lines. The levels of D-glucose and D-glucose-6-phosphate were also relatively similar between two lines. Moreover, the levels of phosphoenolpyruvic acid in both lines were highly increased upon *U. maydis* infection. Three aromatic amino acids, namely phenylalanine, tryptophan, and tyrosine, remained unchanged upon infection in both lines, whereas salicylic acid, caffeic acid, ferulic acid, and sinapic acid, which belong to the phenylpropanoid biosynthesis pathway, were accumulated upon *U. maydis* infection (Figure 5A). Intermediates of these processes, including cinnamic acid and p-coumaric acid, did not show obvious change upon *U. maydis* infection, while shikimic acid, one of the most important metabolites associated with the glycolysis, amino acids, and phenylpropanoid metabolism pathways in plants, was obviously strongly accumulated in mock samples of resistant line CML326 and up-regulated in susceptible line GEMS15 but decreased in resistant line in response to *U. maydis* inoculation (Figure 5A,B), revealing the potential involvement of shikimic acid in the defense response to *U. maydis*.

For genes involved in the above-mentioned metabolism processes, *DAHP* (*phospho-2-dehydro-3-deoxyheptonate aldolase*), *Shikimate dehydrogenase*, and *enolpyruvyl shikimate phosphate synthase* were induced upon *U. maydis* inoculation, among which gene *103637516* (*DAHP*) was specifically up-regulated in CML326 but not GEMS15 in response to *U. maydis* (Figure 5A), supporting the higher levels of shikimic acid in CML326. In addition, gene *100191326* encoding shikimate kinase was induced, while expression of two other shikimate kinase genes, *100283991* and *100191176,* was repressed. Furthermore, genes encoding chorismate synthase, chorismite mutase, phenylalanine ammonia-lyase, cinnamic acid 4-hydroxylase, and chalcone synthase were significantly induced, indicating the positive correlation of phenylpropanoid and flavonoid biosynthesis with their enhanced gene expression levels. Furthermore, gene *542464* encoding cinnamoyl CoA reductase was induced, while gene *100125646* encoding caffeic acid 3-O-methyltransferase and *542663* encoding cinnamyl alcohol dehydrogenase were down-regulated, suggesting the dynamic and complex regulation of lignin biosynthesis in maize during *U. maydis* infection (Figure 5A and Appendix A).

## 4. Discussion

### 4.1. Transcription and Metabolism Reprogramming Networks Regulate Maize Resistance to Corn Common Smut

Plant defense response to pathogens is usually tightly controlled at transcriptional, translational, and metabolism levels. The typical gene-for-gene interaction mechanism, usually identified in plant-biotrophic pathogen, has not been found during maize–*Ustilago* interaction, which is therefore considered to be a quantitative disease resistance (QDR) [20]. In this study, time-course transcriptome analysis using a pair of contrast lines showed that numerous maize genes were uniquely up-regulated or down-regulated during maize in response to *U. maydis* infection. Furthermore, WGCNA analysis identified 993 DEGs that were either commonly or specifically expressed in both lines, and the large portion of DEGS specifically induced by *U. maydis* were related to plant defense to fugal, ROS regulation, and amino acid metabolism regulation, suggesting that this trait is likely controlled by multiple genes and complex mechanisms.

In order to characterize the candidate genes associated with resistance to *U. maydis*, combined transcriptome and metabolome analysis were conducted. This showed that host regulation of defense response, photosynthesis, and cell cycle could play important roles in the interaction between *U. maydis* and maize. Co-expression analysis revealed that defense response genes were clustered in particular modules involving multiple metabolism pathways, including ROS, amino acids, and lipids, indicating that metabolism reprogramming was highly associated with defense response of maize to *U. maydis*. Nitrogen allocation has been reported in maize seedlings upon *U. maydis* inoculation [30], while the relationship between amino acids and defense response against *U. maydis* has not been explored [17]. In addition, cell-cycle-related genes were greatly repressed in CML326, but not in GEMS15, at early time points, e.g., 1 and 2 dpi upon infection, which is in line with previous reports that *U. maydis* infection could activate host defense at an early stage but induce growth at later stage [18,31,32,33]. All these results together suggest that *U. maydis* infection could activate maize transcription reprogramming and regulate the host metabolism process, thereby dynamically modulating maize resistance to *U. maydis* during their biotrophic interaction.

### 4.2. Amino Acid and Phenylpropanoid Metabolism Are Associated with Defense Response to U. maydis

It has been well-documented that phenylpropanoid metabolism plays important roles in plant defense response to pathogens [34], among which phenylpropanoid-related compounds, such as salicylic acid, lignin, and anthocyanin have been reported to accumulate in maize’s response to *U. maydis* [32,35,36]. Specifically, salicylic acid was essential for the plant defense against biotrophic pathogen, whose biosynthesis could be interfered with by *U. maydis* effector Cmu1, a chorismite mutase, resulting in the attenuated SA activity on activation of host immunity [37]. Moreover, as one of the typical phenotypes of host plants upon *U. maydis* infection, anthocyanin could be targeted and manipulated by a secreted effector protein Tin2 of *U. maydis* to facilitate its virulence [36,38]. However, other secondary metabolites participating in maize defense response to *U. maydis* have been less explored.

In this study, phenylpropanoid and flavonoids metabolites were found to accumulate upon *U. maydis* infection, while the content of aromatic amino acids remained unchanged, indicating the enhanced flux in aromatic amino acid metabolism pathways. Moreover, KEGG pathway enrichment clearly showed that genes involved in amino acids and phenylpropanoid metabolism pathways were co-expressed with defense response genes. Specifically, as a key content of glycolysis, amino acid, and phenylpropanoid biosynthesis pathways [39], the content of shikimic acid in susceptible samples was increased upon *U. maydis* infection, which is consistent with results in susceptible maize line Early Golden Bantam [30]. On the contrary, while shikimic acid content in control samples of resistant line was higher, it decreased upon *U. maydis* inoculation. The fact that numerous phenolic acids and flavonoids were accumulated upon *U. maydis* inoculation suggested that the highly accumulated levels of shikimic acid in the resistant line might contribute to the rapid phenylpropanoid biosynthesis and thus the high basal disease resistance levels.

### 4.3. Possible Role of Oxygen Metabolism in Maize Resistance to U. maydis

ROS has been demonstrated to function as an essential component in the defense response in maize to *U. maydis* as a mutation of *U. maydis* genes related to ROS response resulted in defected virulence [14,40,41]. Moreover, the *U. maydis*-secreted effector pep1 could suppress maize peroxidase POX12 to reversely promote the virulence. However, it is not fully understood how host ROS metabolism interferes *U. maydis* infection.

ROS accumulation in host plant could suppress the growth of pathogen, leading to host cell death, thus limiting the infection area. Previous studies on maize–*U. maydis* interaction were mostly performed on a susceptible maize line Early Golden Bantam, in which host cell death were likely repressed; therefore, limited death of plants could be observed [30]. In this study, several maize lines showed a distinct high percentage of death in plants, such as CIMBL80 and CIMBL126, suggesting a positive connection of ROS production to plant cell death. Meanwhile, we identified continuously induced genes and accumulated metabolites related to ROS metabolism upon *U. maydis* inoculation, which might contribute to the induction of ROS accumulation and cell death in plants [42]. Although plant death was often determined as indicating the highest disease severity in maize seedling inoculated with *U. maydis* in multiple studies, early and rapid death in host plants likely indicated unsuccessful colonization of *U. maydis.* It remains to be further investigated whether the impact of cell death on host resistance to *U. maydis* is indeed line-specific, and mechanisms underlying how ROS-related genes regulate the resistance also need to be further investigated.

In summary, through joint analysis of comparative transcriptome and metabolome using a pair of lines with contrast disease phenotype to *U. maydis*, we showed that *U. maydis* infection could activate host transcriptional and metabolism reprogramming to subsequently regulate maize resistance to *U. maydis*. In particular, a series of metabolites related to amino acid, phenylpropanoid, and flavonoid biosynthesis were activated during *U. maydis*–maize interaction, among which shikimic acid seemed to play essential roles in the basal resistance of maize to *U. maydis*. Therefore, this study could provide evidence for the genetic and molecular mechanisms underlying the complexity of common smut resistance in corn at both transcriptional and metabolism levels.

## Figures and Tables

**Figure 1 genes-12-01789-f001:**
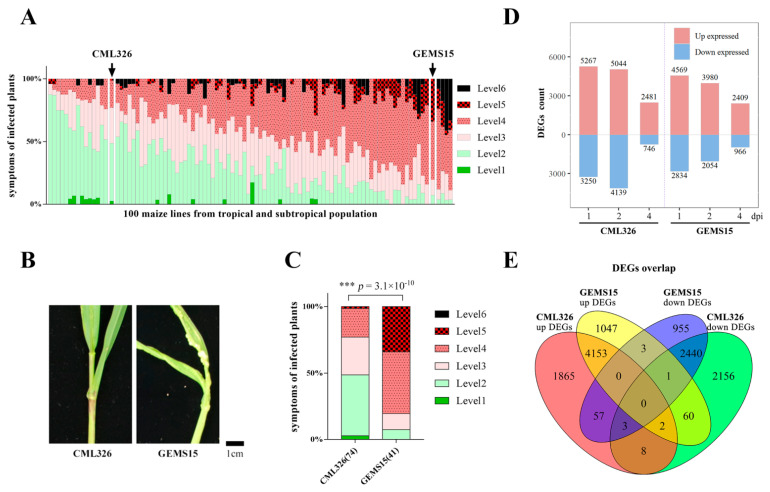
Transcriptome profile of resistant and susceptible maize lines upon *U. maydis* infection. (**A**) Disease symptom distribution among 100 different maize lines. (**B**) Disease symptoms of resistant and susceptible maize lines at 8 days post infection with *U. maydis.* (**C**) Disease index distribution in resistant and susceptible maize lines at 8 days post infection with *U. maydis*. The number of injected plants was labeled in the brackets, and *** indicated significant different distribution in Kruskal–Wallis test *p* < 0.001. (**D**) Number of DEGs in transcriptome profiling of *U. maydis* infected samples relative to mock samples. (**E**) Venn diagram showing overlap of DEGs in resistant and susceptible samples.

**Figure 2 genes-12-01789-f002:**
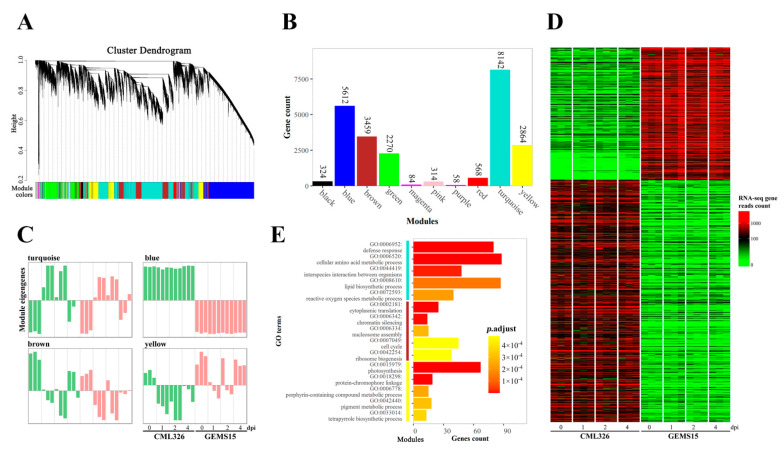
WGCNA and KEGG analysis of DEGs. (**A**) Cluster dendrogram of genes identified in transcriptomic analysis. (**B**) Number of genes in different WGCNA modules. (**C**) Eigengenes of turquoise, blue, brown, and yellow modules. (**D**) Heatmap of genes specifically highly expressed in CML326 or GEMS15 in the blue module. (**E**) Top 5 biological processes enrichment of high module membership genes in turquoise, brown, and yellow modules.

**Figure 3 genes-12-01789-f003:**
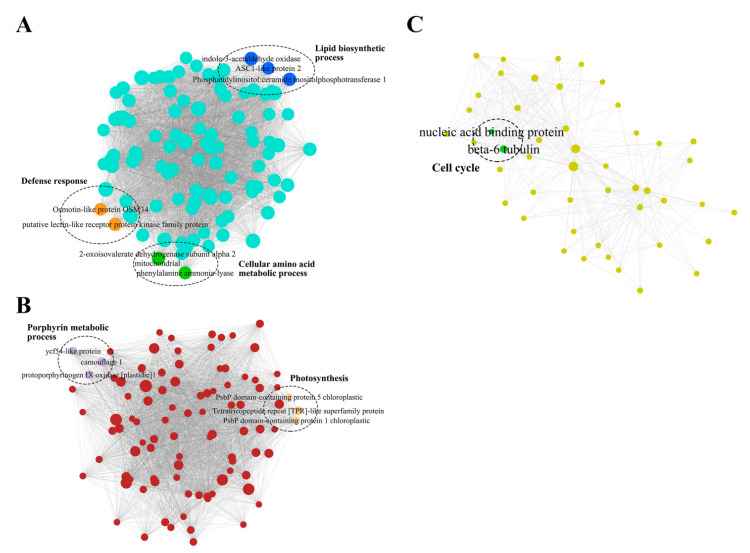
Co-expression network of top 100 co-expressed genes in (**A**) turquoise, (**B**) brown, and (**C**) yellow modules. Dots in network represent genes, lines in network represent co-expression linkage, and dot size indicates strength of co-expression connective within module. Colored dots in dashed ellipse were genes’ annotation in module GO enriched biological process.

**Figure 4 genes-12-01789-f004:**
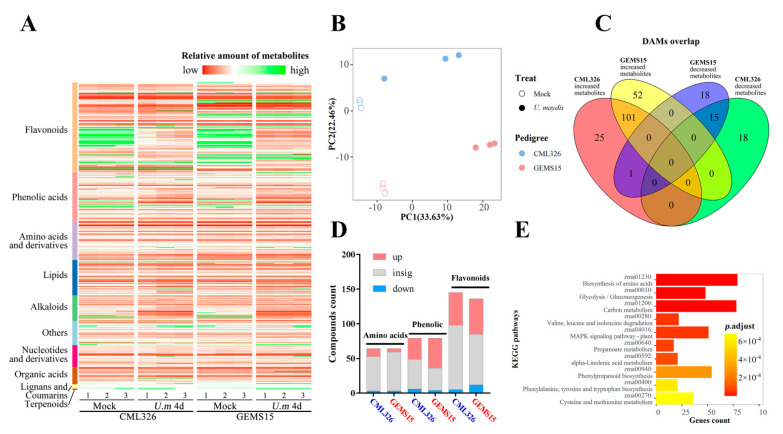
Metabolomic profiling of resistant and susceptible maize lines upon *U. maydis* infection. (**A**) Contents of metabolites detected in metabolome of CML326 and GEMS15 upon *U. maydis* infection. (**B**) PCA analysis of samples in metabolomics profiler. (**C**) Overlapping of significantly changed metabolism content in resistant and susceptible samples infected with *U. maydis* relative to mock samples. (**D**) Statistics of compounds in amino acids and derivatives, phenolic acids, and flavonoids that showed significant change in content upon infection with *U. maydis* at 4 days compared with mock. (**E**) Top 10 KEGG pathways with enrichment of hub genes in turquoise module.

**Figure 5 genes-12-01789-f005:**
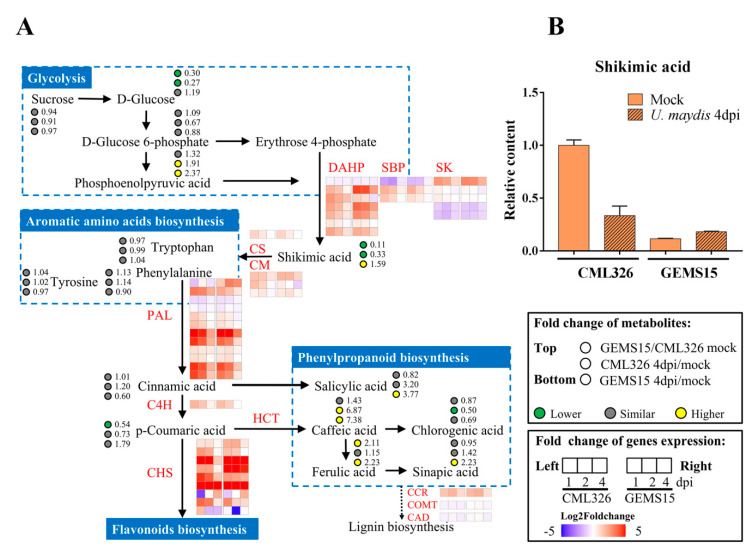
Dynamic changes of gene expression levels and metabolites contents in shikimic acid pathways. (**A**) Pathways involved in glycolysis, aromatic amino acids biosynthesis, phenylpropanoid biosynthesis, and flavonoids biosynthesis and regulation of related genes and metabolites in CML326 and GEMS15 in response to *U. maydis* inoculation. The diagrams in the squares on the right indicate the treatments, time course for metabolite changes, and gene expression levels. DAHP: phospho-2-dehydro-3-deoxyheptonate aldolase; SBP: shikimate biosynthesis protein; SK: shikimate kinase; CS: chorismate synthase; CM: chorismite mutase; PAL: phenylalanine ammonia-lyase; C4H: cinnamic acid 4-hydroxylase; CHS: chalcone synthase; HCT: hydroxycinnamoyl-CoA shikimate/quinate hydroxycinnamoyltranferase; CCR: cinnamoyl CoA reductase; COMT: caffeic acid 3-O-methyltransferase; CAD: cinnamyl alcohol dehydrogenase. (**B**) Relative contents of shikimic acid in CML326 and GEMS15 samples upon infection with *U. maydis*.

## Data Availability

Sequence-based gene count datasets generated from this study have been deposited to NCBI Gene Expression Omnibus (GEO) under the accession number GSE172071 (https://www.ncbi.nlm.nih.gov/geo/query/acc.cgi?acc=GSE172071, accessed on 15 April 2021).

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
