# Peer review of "Dissection of the Complex Transcription and Metabolism Regulation Networks Associated with Maize Resistance to Ustilago maydis"

_genes, 2021, doi:10.3390/genes12111789_

Round 1

Reviewer 1 Report

Congratulations on the study. However, in the case of this paper, I would have a recommendation that it would increase its value. It would be good to include one or two conclusions and not limit yourself only to discussions.

Author Response

A: thanks a lot for your constructive suggestion! We have added one short paragraph in the end of discussion to conclude our work. Please refer lines 454-462 in the new version.

Reviewer 2 Report

The study was focused on dissection of the complex transcription and metabolism regulation networks associated with maize resistance to Ustilago maydis. The Authors performed comparative transcriptome and metabolome investigations on a pair of maize lines with contrast resistance to U. maydis post infection. WGCNA of transcriptome profiling reveals that defense response, photosynthesis, and cell cycle are critical processes in maize response to U. maydis, and metabolism regulation of glycolysis, amino acids, phenylpropanoid, and reactive oxygen species are closely correlated with defense response. Metabolomic analysis supported that phenylpropanoid and flavonoids biosynthesis were induced upon U. maydis infection, and obviously high content of shikimic acid, a key compound in glycolysis and aromatic amino acids biosynthesis pathways, was detected in resistance samples.

In my opinion, the manuscript is well organized and presents interesting research data. However, I recommend the following minor revisions:

  • The Authors used SYBR Green fluorescent dye during gene expression studies, hence, it is obligatory to perform Melting Curve Analysis, and results of this examination should be added in the manuscript or Supplementary file (e.g., JPG or TIFF file),
  • I recommend including the electropherograms presenting the RNA bands in agarose gels in the manuscript or in the Supplementary file – it would provide information regarding quality of RNA samples,
  • RIN = RNA Integrity Number numbers of the RNA samples should be presented in the manuscript,
  • Citation of 2-ΔΔCT method should be added in the References,
  • Moderate English changes are required.

Author Response

In my opinion, the manuscript is well organized and presents interesting research data. However, I recommend the following minor revisions:

  • The Authors used SYBR Green fluorescent dye during gene expression studies, hence, it is obligatory to perform Melting Curve Analysis, and results of this examination should be added in the manuscript or Supplementary file (e.g., JPG or TIFF file),

A: Thanks a lot for your suggestion! We have supplemented the additional figure of melting curve analysis for qRT-PCR. Please refer Supplementary Figure S4 and relative description for methodology on line 139, and results were also provided on line 195-106.

  • I recommend including the electropherograms presenting the RNA bands in agarose gels in the manuscript or in the Supplementary file – it would provide information regarding quality of RNA samples,

A: Following your suggestion, we included an electropherograms as Supplementary Figure S2, showing all RNA samples in this study.

  • RIN = RNA Integrity Number numbers of the RNA samples should be presented in the manuscript,

A: Thank you so much for your suggestion! The RIN values of all RNA samples were presented in new Supplementary Table 2. Also, the corresponding methods about quality control of RNA samples were provided on line 110-113.

  • Citation of 2-ΔΔCT method should be added in the References,

A: The new citation for 2-ΔΔCT method was added in the reference list (reference #28).

  • Moderate English changes are required.

A: We appreciate very much your constructive comments We have looked through entire manuscript carefully and made revisions on English language/wording accordingly.